# Recreational drug use among Nigerian university students: Prevalence, correlates and frequency of use

**Anthony Idowu Ajayi**[1]*, **Oluwaseyi Dolapo Somefun**[2]

**1** Population Dynamics and Sexual and Reproductive Health and Rights Unit, African Population and Health Research Centre, Manga Close, Nairobi, Kenya, **2** Demography and Population Studies (DPS), University of the Witwatersrand, Johannesburg, South Africa

\* ajayianthony@gmail.com, aajayi@aphrc.org

**Data Availability Statement:** Data underlying the study is from S1 Dataset of the published article DOI: 10.1371/journal.pone.0221804 (Dataset for understanding the predictors of condom use self-efficacy among university students in Nigeria;

## Abstract

### Background

Given the paucity of data on recreational drug use and the recent media attention on the abuse of drugs such as codeine cough syrups and tramadol, in Nigeria, our study examined the prevalence and frequency of recreational drug use among young adults from two Nigerian universities. We drew from the Socio-ecological Model to examine the influence of factors at the individual and family level on recreational drug use among adolescents and young adults.

### Methods

This cross-sectional study was conducted between February and March 2018 among a final sample of 784 male and female university students selected using stratified random sampling. Binary logistic regression was used to identify significant predictors of ever use and current use of drugs.

### Results

Our analyses showed that 24.5% of students had ever used drugs for recreational purposes, and 17.5% are current users. The median drug use frequency over the past month was six days among current users (n = 137). In the multivariable analyses, living in the same household as one's mother (AOR 0.28 95% CI 0.16–0.49), adequate family support (AOR 0.48 95% CI 0.26–0.89) and frequent attendance of religious fellowships (AOR 0.13 95% CI 0.07–0.25) were significantly associated with a lower likelihood of recreational drug use. However, male sex (AOR 1.52 95% CI 1.05–2.21) was associated with higher odds of recreational drug use.

### Conclusion

The family should be considered as an important unit to sensitize young people on the harmful effects of drug use. It is also vital that religious leaders speak against drug use in their

https://doi.org/10.1371/journal.pone.0221804.
s002).

**Funding:** The authors received no specific funding
for this work.

**Competing interests:** The authors have declared
that no competing interests exist.

various fellowships. There is a need to address recreational drug use on Nigerian campuses
by educating students about its adverse impacts.

## Background

The illicit use of drugs for recreational purposes or for eliciting intoxicating effects has been
recognised to be a growing and burdensome public health issue particularly among young
adults in high-income countries [1–3] and recently in developing countries [4–7]. The global
deaths caused directly by the use of illicit drugs have increased by 60% from 105,000 deaths in
the year 2000 to 168,000 deaths in 2018 [8]. The adverse effects of drug abuse include non-
communicable diseases, cardiovascular and central nervous system collapse [9], addiction
[10], mental health issues [11, 12], accidents [13], involvement in criminal activity [14] and
risky sexual behaviours [15, 16]. Also, mental health disorder among youths aged 10–24,
which accounts for about 2% of global deaths, has been linked to illicit drug use [17]. Cannabis
use can affect school performance [18], especially in mathematics. Heavy and regular mari-
juana use during teenage years can result in an 8-point reduction in Intelligence Quotient (IQ)
[19].

According to the United Nations Office on Drugs and Crime (UNODC), Nigeria is one of
the current highest consumer of cannabis and amphetamine in Africa [8]. Nevertheless, stud-
ies on recreation drug use in Nigeria are scarce and exitsing studies have focused on high
school students and mostly on the use of alcohol. Also, most of the studies focus on the role of
individual-level factors and peer influence. There is, however, a paucity of studies examining
the influence of family/household-level and social factors on recreational drug use among ado-
lescents and young adults in Nigeria. Our study fills this gap by examining the drug use preva-
lence and frequency as well as the role of family/household-level and social factors on
recreational drug use among adolescents and young adults.

### Theoretical underpinning

We drew from the socio-ecological model (SEM) [20] to understand the role of family/house-
hold-level and social factors on recreational drug use among adolescents young adults. The
SEM has proven to be an instrumental theoretical framework for addressing several youth
development outcomes. The SEM posits that youth behaviours are not influenced only by
their individual (intrapersonal) characteristics but by other factors in their environment which
could be at the family, school, peer, community and national level. This model assumes that
interactions between youth and different levels are mutual as they both influence each other.

The socioecological model postulates that the family is a child's early microsystem for learn-
ing how to live and about the real world. The influence of the family is strong in the life of a
child. It provides the nurturing centrepiece for the child. One mechanism through which the
family influences young peoples drug use may be through living arrangements and parental
monitoring. The age of adolescence is a transition phase, and the presence of parents leaves a
permanent influence that is crucial for the development of youth. The family instils norms and
values in a child through constant training, control and monitoring, which is also dependent
on the household living arrangement. The family norms and values instilled from childhood
to adolescence period will continue to influence the behaviour of young people even until their
adulthood. There is evidence that youths with a higher level of parental monitoring have lower
odds for risky behaviours [21].

Based on SEM, family structure, living arrangement, and family support could influence young people's use of drugs. A study has shown that higher parental education and income are associated with marijuana use in the United States [22]. We, therefore, posit that adequate family support will be associated with reduced odds of recreational drug use among adolescent and young adults in Nigeria.

Living in one's parent household may influence the amount of time spent with the child and may also influence closeness to the child that may continue even when the youth leaves the home. Individuals who did not live with their parents due to deaths or divorce may have missed out on some of the critical parental lessons capable of shaping their later life behaviours. The effect of living in the same household as one's parent on drug use has never been investigated in the Nigerian context. Our study proposed that living in the same household as one's parent will reduce the odds of drug use among young people even when they are in universities.

Religion is also a microsystem factor that could impact illicit drug use among adolescents and young adults. Religion teaches codes of ethical behaviours and forbids the use of illicit drugs. As such, we posit that frequent attendance of religious fellowships will be associated with a lower likelihood of illicit drug use among young Nigerian adults in Nigeria.

The socioecological model also illustrates the impact of meso, exo and macrosystems on the development of a child. However, our focus in this study is limited to the impact of microsystem factors such as the family and religion in understanding young people's illicit drug use.

## Methods

### Study design

The data analysed in this study came from a more extensive study, which assessed the sexual health of university students in Nigeria. Full details of the methodology have been published elsewhere [23–25]. This descriptive cross-sectional study was conducted among male and female university students in two Nigerian universities between February and April 2018. The two universities, one owned by the state government and the other owned by the federal government, were selected purposively. A self-designed questionnaire was administered to 800 consenting male and female students through face-to-face interviews. A pilot study was conducted among 20 demographically matched students in another university, and feedback was obtained to improve the questionnaire. Experienced research assistants who were trained purposively for this study conducted the interviews.

### Participants and sampling

The study participants were male and female students of the two selected universities. The population of students in both universities was about 45,000. Participants were selected using stratified random sampling. Stratification was based on sex, level of study, and course of study. Only undergraduate students were included in the study. For representativeness, the estimated sample size was 384 students per university. This was determined using the sample size calculator, at a 95% confidence level, ±5 margin of error. However, 400 students were selected per university after adjusting for possible incomplete responses. Overall, 800 male and female students took part in the study; however, only 784 questionnaires were returned with complete responses.

### Ethical statement

All participants provided written consent, and the rights of participants to privacy, anonymity, and confidentiality were maintained throughout the study. The University of Fort Hare and

Ondo State ethical review committees approved the study protocol. We provided students under age 18 years with an additional parental/guardian consent form to complete. The students were given a week to obtain their parent/guardian's consent to participate in the study. About 56 under-18 students who took part in the study obtained their parent or guardian's consent and also assented to participate in the study.

## Dependent variables

The main outcome variable was a two-category (yes/no) nominal measure of recreational drug use. Participants were asked: "Have you ever used substances/drugs like Codeine, Marijuana, Tramadol for pleasure, or to ease tension/stress?" To estimate the recent use of drugs for recreation, we asked participants: "Do you currently use substances/drugs like Codeine, Marijuana, Tramadol for pleasure, or to ease tension/stress? To understand the frequency of recreational drug use, we asked participants to state the number of days they have used recreational drugs in the past 30 days. We did not ask questions on specific drugs, which has been noted as a limitation of this study.

## Independent variables

Our variable selection was informed by the SEM [20]. We included individual-level characteristics, such as age and gender. Age was measured by asking participants to indicate their age at their last birthday. We later categorised these ages into a categorical variable for bivariate and multivariable analyses. Also, participants were asked to indicate their gender.

Based on the SEM propositions, family/household factors are important microsystem predictors of health outcomes [20]. We included several family factors, such as family structure, family support, death of parents, and living with parents in this study. Family structure was measured by asking participants to describe their family type. We provided four categories for participants to choose from, which include nuclear family (two-parent family), single-parent family, polygamous family (mother having a co-wife), and foster family (residing with uncles, aunties, grandmother or father).

Family support was measured by asking participants to rate the level of support they receive from their parents. Responses were categorised as adequate support, moderate support, insufficient support, and no support. Our operationalisation of family support substituted for family wealth. We note that it is perhaps not only the amount of money a child receives from home that matters but also how the child perceived the money to be sufficient to meet their needs or not. As such, we asked participants to consider the support they get from their family in totality, that is, including support beyond money. Also, we used the term family support loosely given that some students were orphans. Thus, they could reference the support they received from their guardians as family support. Financial support is important for students, given that a majority of them depend heavily on their parents for university tuition fee and other living expenses. Thus, the lack of financial support means that students feel unsupported. However, it is important to note that in rare cases, parents could provide adequate financial support but less emotional support. Overall, our question is mostly an indication of how participants perceived the level of support they receive from their parents. Rather than a measure of monetary value, perceived parental support is an indication of the value students attribute to the support (both financial and beyond) they receive from their parents.

We also asked participants whether their parents are alive and if they live with their parents when not in school. Responses were a binary choice of "yes" or "no".

Another microsystem factor we considered in this study that could potentially influence young people's drug use for recreational drug use is religion. We asked participants to indicate

their religion from a list containing the main religious groups in Nigeria. Participants' religious affiliation was classified as Christian- Orthodox, Christian-Pentecostal, and Muslim. We further asked participants about their frequency of attendance of religious fellowships. We provide five options, which include every day, twice a week, once a week, once a month, and once a year. We classified those who attended religious gatherings every day and at least twice a week as frequent attendants, those who attended once a week as moderate attendants, and those who attended once a month or in a year as low attendants.

## Statistical analysis

The data generated through the questionnaires were coded and captured into the Statistical Package for the Social Sciences (IBM SPSS Statistics Version 24). Data were cleaned for possible data entry errors. Descriptive statistics were used to summarise all variables of interest. Frequency counts and percentages were computed. Mean, median, and standard deviation were calculated for continuous variables. Unadjusted binary logistic regression model was used to examine the effect of each individual level factors, family/hosuehold factors, and religion on having ever used drugs and current use of drugs for recreational purposes. Adjusted binary logistic regression model was used to examine the predictors of drug use among the students. 95% confidence intervals were estimated for each odds ratio. All $p$-values less than 0.05 were considered statistically significant.

# Results

The median age of study participants was 22 years. The demographic characteristics of the study participants are presented in Table 1. Most participants were aged 24 years and below (66.9%), resided off campus (80.1%), from a nuclear family (58.2%), and received adequate support from home (70.9%).

## Recreational drug use

About a quarter of the students had ever used substances/drugs like Codeine, Marijuana, and Tramadol for recreational purposes and the proportion varied by age (16–19 years; 18.0%, vs 20–36 years; 27%), sex (male; 29.6%. vs female; 19.2%), and family support (two-parent family 19.7% vs other family types 30.8%).

The results of the unadjusted and adjusted logistic regression are presented in Table 2. In the unadjusted logistic regression model, sex, age, living with the father, mother alive, living with the mother, family structure, family support, religious affiliation and higher frequency of religious attendance (at least twice a week) were significantly associated with having ever used drugs for recreational purpose. In the adjusted model, only being a male, living with the mother, family support, Christian -orthodox religious affiliation, and frequent attendants of religious fellowships remained significantly associated with having ever used drugs for recreational purposes. Students who live in the same household as their mothers had lower odds of having ever used drugs for recreational purposes compared with those who did not. Students who receive adequate family support were 52% less likely to have ever used drugs for recreational purposes compared with those who received insufficient or no support. Christian-orthodox religious affiliation was associated with a lower likelihood of drug use for recreational purposes, with students who attended orthodox churches being 43% less likely to use drugs for recreational purposes compared with those who were Muslims. Also, those who attended religious services at least twice a week were 87% less likely to have ever used drugs for recreational purposes compared with those who attended only once a month or a year.

**Table 1. Sociodemographic and family characteristics of study participants by gender.**

| Variables | All participants | Male | Female |
|---|---|---|---|
| All participants | 784 (100) | 402 (51.3) | 382 (48.7) |
| Age | | | |
| Below 20 years | 219 (27.9) | 76 (18.9) | 143 (37.4) |
| 20–24 years | 384 (49.0) | 207 (51.5) | 177 (46.3) |
| Above 24 years | 181 (23.1) | 119 (29.6) | 62 (16.2) |
| Residence Type | | | |
| University residence | 156 (19.9) | 61 (15.2) | 95 (24.9) |
| Off campus residence | 627 (80.1) | 340 (84.8) | 287 (75.1) |
| Living arrangement | | | |
| I live alone | 237 (30.7) | 116 (29.4) | 121 (31.9) |
| Live with one room mate | 315 (40.8) | 161 (40.9) | 154 (40.6) |
| Have more than one roommate | 221 (28.6) | 117 (29.7) | 104 (27.4) |
| Religious Background | | | |
| Christian orthodox | 304 (38.9) | 166 (41.3) | 138 (36.3) |
| Christian Pentecostal | 270 (34.5) | 131 (32.6) | 139 (36.6) |
| Muslim | 208 (26.6) | 105 (26.6) | 103 (27.1) |
| Father alive | | | |
| Yes | 654 (83.4) | 328 (81.6) | 326 (85.3) |
| No | 130 (16.6) | 74 (18.4) | 56 (14.7) |
| Live in the same household as your father | | | |
| Yes | 567 (82.3) | 277 (78.9) | 290 (85.8) |
| No | 122 (17.7) | 74 (21.1) | 48 (14.2) |
| Mother alive | | | |
| Yes | 700 (89.5) | 353 (88.0) | 347 (91.1) |
| No | 82 (10.5) | 48 (12.0) | 34 (8.9) |
| Live in the same household as your mother | | | |
| Yes | 642 (89.4) | 315 (85.8) | 327 (93.2) |
| No | 76 (10.6) | 52 (14.2) | 24 (6.8) |
| Family type | | | |
| Single-parent family | 199 (25.5) | 124 (31.0) | 75 (19.7) |
| Nuclear family | 454 (58.2) | 212 (53.0) | 242 (63.7) |
| Polygamous family | 90 (11.5) | 49 (12.3) | 41 (10.8) |
| Foster family | 37 (4.7) | 15 (3.8) | 22 (5.8) |
| Family support | | | |
| Adequate | 555 (70.9) | 263 (65.6) | 292 (76.4) |
| Moderate | 167 (21.3) | 105 (26.2) | 62 (16.2) |
| Insufficient | 44 (5.6) | 24 (6.0) | 20 (5.2) |
| No support | 17 (2.2) | 9 (2.2) | 8 (2.1) |

## Current drug use

The prevalence of current drug use for recreational purposes was 17.5%, with significant sex, age, and family characteristics variations. In the unadjusted model presented in Table 3, sex, age, living in the same household as one's father, mother alive, living in the same household as one's mother, family structure, family support, religious affiliation and higher frequency of religious attendance were associated with recreational drug use. However, age, living in the same household as one's mother, family support, and frequent religious fellowship attendance

**Table 2. Multivariable analysis showing predictors of ever use drugs for recreational purposes among university students.**

| Variables | Yes n (%) | No n (%) | Unadjusted Odds Ratio (95% CI) | Adjusted Odds Ratio (95% CI) |
|---|---|---|---|---|
| All participants | 191 (24.5) | 589 (75.5) | | |
| Gender | | | | |
| Male | 118 (29.6) | 281 (70.4) | 1.77 (1.27–2.47)* | 1.52 (1.05–2.21)* |
| Female (Ref) | 73 (19.2) | 308 (80.8) | 1 | 1 |
| Age | | | | |
| Below 20 | 39 (18.0) | 178 (82.0) | 0.59 (0.40–0.88)* | 0.69 (0.44–1.07) |
| 20 and above (Ref) | 152 (27.0) | 411 (73.0) | 1 | 1 |
| Father alive | | | | |
| Yes | 153 (23.5) | 498 (76.5) | 0.74 (0.48–1.12) | 0.97 (0.54–1.77) |
| No (Ref) | 38 (29.5) | 91 (70.5) | 1 | 1 |
| Live in the same household as father | | | | |
| Yes | 123 (21.8) | 442 (78.2) | 0.43 (0.28–0.65)*** | 0.77 (0.47–1.29) |
| No (Ref) | 48 (39.3) | 74 (60.7) | 1 | 1 |
| Mother Alive | | | | |
| Yes | 157 (22.5) | 540 (77.5) | 0.42 (0.26–0.68)*** | 1.77 (0.88–3.57) |
| No (Ref) | 33 (40.7) | 48 (59.3) | 1 | 1 |
| Live in the same household as mother | | | | |
| Yes | 125 (19.5) | 515 (80.5) | 0.18 (0.11–0.30)*** | 0.28 (0.16–0.49)*** |
| No (Ref) | 43 (57.3) | 32 (42.7) | 1 | 1 |
| Family Types | | | | |
| Nuclear | 89 (19.7) | 362 (80.3) | 0.55 (0.40–0.77)*** | 0.90 (0.58–1.38) |
| Single parent/polygamous/foster care (Ref) | 100 (30.8) | 225 (69.2) | 1 | 1 |
| Family Support | | | | |
| Adequate/moderate support | 160 (22.0) | 559 (77.7) | 0.29 (0.17–0.49)*** | 0.48 (0.26–0.89)* |
| No or insufficient support (Ref) | 30 (50.0) | 30 (50.0) | 1 | 1 |
| Religious affiliation | | | | |
| Christian-Orthodox | 66 (21.7) | 238 (78.3) | 0.63 (0.42–0.95)* | 0.57 (0.37–0.90)* |
| Christian-Pentecostal | 62 (23.0) | 208 (77.0) | 0.68 (0.45–1.02) | 0.74 (0.47–1.15) |
| Muslim | 64 (30.5) | 146 (69.5) | 1 | 1 |
| Frequency of religious attendance | | | | |
| Frequent attendants (At least twice a week) | 89 (17.4) | 423 (82.6) | 0.11 (0.06–0.19)*** | 0.13 (0.07–0.25)*** |
| Moderate attendants (Once a week) | 67 (30.7) | 151 (69.3) | 0.22 (0.12–0.42) *** | 0.30 (0.15–0.60) * |
| Low attendants (One month or in a year) | 36 (66.7) | 18 (33.3) | 1 | 1 |

Ref-reference category

*P-value <0.05

***P-value <0.001, CI-confidence interval

remained significantly associated with current drug use for recreational purposes in the adjusted model. Adequate family support was protective against recreational drug use. Individuals who received adequate support from home were 53% less likely to currently use drugs for recreational purposes compared with those who did not. Students who lived in the same household as their mother were 71% less likely to currently use recreational drugs compared with their counterparts who did not. Students who fellowshipped in orthodox churches were 44% less likely to use drugs for recreational purposes compared to those who were Muslims. Also, those who attended religious services every day or at least twice a week were 86% less

**Table 3. Multivariable analysis showing predictors of current drug use among university students.**

| Variables | Yes n (%) | No n (%) | Unadjusted Odds Ratio (95% CI) | Adjusted Odds Ratio (95% CI) |
|---|---|---|---|---|
| All | 137 (17.5) | 647 (82.5) | | |
| Sex | | | | |
| Male | 84 (20.9) | 318 (79.1) | 1.64 (1.13–2.39)* | 1.39 (0.92–2.12) |
| Female (Ref) | 53 (13.9) | 329 (86.1) | 1 | 1 |
| Age | | | | |
| Below 20 years | 25 (11.4) | 194 (88.6) | 0.52 (0.33–0.83)* | 0.60 (0.36–1.01) |
| 20 years and above (Ref) | 112 (19.8) | 453 (80.2) | 1 | 1 |
| Father alive | | | | |
| Yes | 109 (16.7) | 545 (83.3) | 0.73 (0.46–1.16) | 0.97 (0.50–1.88) |
| No (Ref) | 28 (21.5) | 102 (78.5) | 1 | 1 |
| Live with you | | | | |
| Yes | 87 (15.3) | 480 (84.7) | 0.40 (0.26–0.63)*** | 0.72 (0.41–1.25) |
| No (Ref) | 38 (31.1) | 84 (68.9) | 1 | 1 |
| Mother alive | | | | |
| Yes | 114 (16.3) | 586 (83.7) | 0.53 (0.31–0.90)* | 2.42 (1.12–5.21)* |
| No (Ref) | 22 (26.5) | 61 (73.5) | 1 | 1 |
| Live in the same household as mother | | | | |
| Yes | 89 (13.9) | 553 (86.1) | 0.22 (0.13–0.37)*** | 0.29 (0.16–0.52)*** |
| No (Ref) | 32 (42.1) | 44 (57.9) | 1 | 1 |
| Family type | | | | |
| Nuclear family | 69 (21.2) | 256 (78.8) | 0.63 (0.44–0.92)* | 1.12 (0.68–1.83) |
| Single parent/polygamous/foster parent (Ref) | 64 (14.3) | 384 (85.7) | 1 | 1 |
| Family support | | | | |
| Adequate/moderate support | 26 (42.6) | 35 (57.4) | 0.24 (0.14–0.42)*** | 0.39 (0.21–0.73)* |
| No or insufficient support (Ref) | 26 (42.6) | 35 (57.4) | 1 | 1 |
| Religious affiliation | | | | |
| Christian-Orthodox | 46 (15.1) | 258 (84.9) | 0.62 (0.39–0.97)* | 0.56 (0.34–0.93)* |
| Christian-Pentecostal | 44 (16.3) | 226 (83.7) | 0.68 (0.43–1.07) | 0.76 (0.46–1.25) |
| Muslim | 47 (22.4) | 163 (77.6) | 1 | 1 |
| Frequency of religious attendance | | | | |
| At least twice a week | 61 (11.9) | 451 (88.1) | 0.11 (0.06–0.20)** | 0.14 (0.07–0.26)*** |
| Once a week | 46 (21.1) | 172 (78.9) | 0.21 (0.11–0.40) *** | 0.29 (0.15–0.57)*** |
| Once month or in a year | 30 (55.6) | 24 (55.6) | 1 | 1 |

Ref-reference category

*p-value <0.05

***p-value <0.00, CI-confidence interval

likely to use drugs for recreational purposes compared with those who attended only once in a month or in a year.

## Frequency of drug use

The findings on the frequency of drug use are shown in Fig 1. The median number of days participants used drugs over the past month was six days. Of the 137 students who currently use drugs, 23.8% used drugs in more than 10 days of the last month. Males (30.%; n = 25/84) were significantly more likely to have used drugs for more than 10 days of last month before the study compared to females (13.3%; n = 7/53).

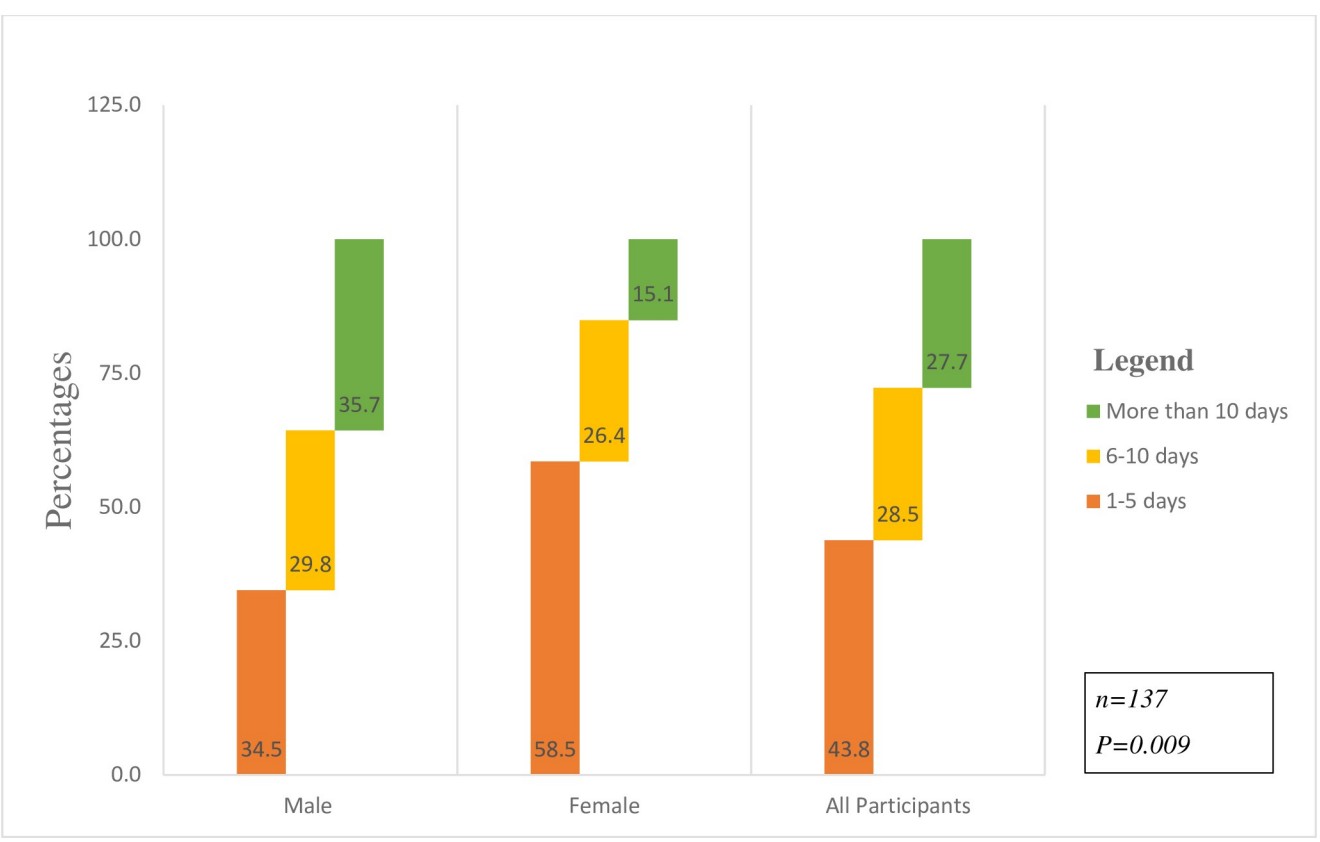

**Fig 1. Frequency of drug use by gender.**

## Discussion

Our study is motivated by the paucity of data, especially among university students, on recreational drug use in Nigeria. Given the media attention to the abuse of drugs, such as Tramadol and Codeine, our study is timely and provides vital data for policymakers tasked with addressing illicit drug use in Nigeria. The present study shows that approximately a quarter of the university students in Nigeria have ever used substances/drugs like Codeine, Marijuana, and Tramadol, among others, while close to one-fifth are current users. The only study conducted among university students in Nigeria [26] is outdated and reported a low prevalence of drug use compared to our study. A study has shown that the prevalence of drug use has increased since the 1990s [8]. Also, the abuse of tramadol and codeine, in particular, may have contributed to the increasing prevalence of recreational drug use among young people in Nigeria. Altogether, there is a need to for policymakers to prioritise Nigeria campuses for drug abuse prevention interventions, considering the deleterious effects of drug use on the health of young people.

Based on SEM [20], factors that influence recreational drug use among young people operate at the individual, microsystem, mesosystem, exosystem and macrosystem levels. From our findings, gender was the only individual-level factor significantly associated with recreational drug use. Males were significantly more likely to use drugs compared to females. Similarly, males are frequent drug users relative to females. These results are consistent with previous studies on drug use [27–29]. One plausible reason for this is that boys are more favourable to taking risks compared with girls. Boys generally engage more in unhealthy lifestyles relative to

girls [30]. Another plausible reason for the observed gender differences in drug use in this study is cultural expectations. There is more societal stigma and shame around women and girls drug use in the Nigeria context [8]. Also, this finding suggests that males are more exposed to the deleterious consequences of drug use compared to females. However, the prevalence of drug use is relatively high in both genders, which suggests the need to target both males and females with any interventions tailored to address illicit drug use on Nigerian campuses.

Also, based on SEM [20], family/household factors are important predictors of recreational drug use. Our study shows that being from a nuclear family was associated with a lower likelihood of drug use in the unadjusted model; however, the effect size was no longer significant after controlling for demographic factors. Previous studies have reported conflicting findings on the relationship between family structure and drug use among young people [31–34]. Some studies show that family structure is associated with drug use [32–34]; however, there is evidence that family structure had little impact on drug use after controlling for covariates [35]. The plausible reason why the nuclear family had some effect on adolescent and young adults' illicit drug use is that having and living with both parents means more parental monitoring, training and control. When one parent is away, the other parent can take over the caring of the child, unlike in a single-parent family.

We also found that living with one's father is protective against drug use in the unadjusted regression analysis, but the effect size reduced and was no longer significant after adjusting for important covariates. A study has shown that the presence of a father is protective against drug use, especially for boys [36]. The presence of fathers is important for adolescents and particularly boys. Our study further shows that living with the mother is protective against recreational drug use. The protective effect of the presence of the mother in the life of young person against drug abuse is substantial and remains after controlling for important covariates. Students who lived in the same household as their mother had a lower likelihood of current and ever use of drugs compared with their counterparts who did not. One explanation for this finding is that young adults who did not live with their mothers, either as a result of death or divorce, may experience behavioural challenges. Based on anecdotal knowledge, mothers are essential for exerting control over the behaviours of young adults, especially in Nigeria, where the burden of parenting lies mainly on mothers. An alternative explanation could be those young adults who did not live with their mother either as a result of death or divorce use drugs as a coping mechanism for dealing with the loss of their mothers. Our explanation is bolstered by studies that established that the death of a mother could devastate the health and economic well-being of a family, especially that of the children [37–40]. Children whose mothers died are abandoned by their fathers, undernourished, forced to drop out of school, to take on difficult household and farm tasks and are far less likely to survive [37–40]. Overall, the importance of having both mother and father present in the life of a young person cannot be overemphasised. By and large, the protective effect of mothers against drug use is far more significant compared with fathers. Our finding is supported by a study which demonstrated that drug use among daughters living with single-fathers exceeded that of daughters living with single-mothers [41].

Another key finding of this study is that adequate family support was protective against recreational drug use. Individuals who received adequate support from home were 53% less likely to currently use drugs for recreational purposes compared towith those who did not. This study is consistent with a previous study that argues that parental support and monitoring are important predictors of young people's health outcomes [42]. Our study provides strong evidence for the importance of family support as a protective factor against illicit drug use. A study has established that young people who have strong bonds with their families are less

likely to use recreational drug [43]. The importance of family support on young people's illicit drug use is noted in a study that shows that high levels of parental monitoring and family support were effective in buffering the relation between witnessing violence with the initiation of cigarette as well as advanced alcohol use at low levels of witnessing violence [44]. However, a study showed that patterns of drug use among young people coincided with patterns of family conflict, but not family support [45].

We also found strong evidence in support of the protective effect of religion on recreational drug use. Frequent association and fellowship with other people tend to serve as social control against illicit drug use. Religion has a strong influence on the lives of young people. It serves an important agent of socialisation and an institution where young people are taught to behave well and follow religious norms. Religious teachings forbid delinquent behaviours; it teaches young people to follow the laws, which prohibits illicit drug use. Religion acts as a form of social control against illicit drug use, and those that regularly worship tend to abstain from illicit drug use.

The limitation of this study includes the use of cross-sectional design, which does not provide information on causal inference. We conducted this study among a subset of Nigerians young people who are more educated compared to others. As such, our study is not generalizable to all young people in Nigeria. Also, we did not ask questions about specific drugs or the use of multiple drugs. As such, we are unable to estimate the prevalence and correlates of use of specific drugs. Further, we did not examine the role of peer influence and other structural factors on recreational drug use in this study. Despite these limitations, our study advances the discourse of recreational drug use in the Nigerian context and identifies key protective factors against drug use. Future studies should estimate the prevalence of the use of specific drugs like Codeine Syrup, Tramadol, and Cannabis as well as the influence of peers on recreational drug use.

## Conclusion

This study estimated the prevalence of and examined the correlates of recreational drug use among Nigerian university students. Our results showed a high prevalence of recreational drug use among students with significant sex variations. Given the study findings and harmful consequences of illicit drug use, there is a need for interventions to address recreational drug use on Nigerian campuses. The SEM provided a framework for understanding the influence of individual-level factors, family/household factors and religion on recreational drug use among adolescent and young adults. Our findings underscore that relevance of SEM in understanding adolescent and young people's behaviours. Even though SEM did not inform the design of this study, it proves relevant in understanding the context in which family factors impact adolescents development. One criticism of SEM is that it is difficult to evaluate all its components empirically. We did not include other important SEM factors, including peer influence and structural factors such as governmental policies and law enforcement in our study. As such, our study did not evaluate the overall applicability of SEM; rather, it drew from SEM to understanding the factors influencing adolescents and young adults' drug use.

## Acknowledgments

The authors would like to acknowledge the research assistants (Ojo Oluwayomi Emmanuel, Ismail Kafayat, and Abdulazeez Abioye) for their contribution during the data collection. AIA would like to express gratitude to the African Population and Health Research Center, which granted him the opportunity to work on this study by granting him a postdoctoral fellowship position.

## Author Contributions

**Conceptualization:** Anthony Idowu Ajayi, Oluwaseyi Dolapo Somefun.

**Data curation:** Anthony Idowu Ajayi.

**Formal analysis:** Anthony Idowu Ajayi.

**Investigation:** Anthony Idowu Ajayi.

**Methodology:** Anthony Idowu Ajayi.

**Supervision:** Anthony Idowu Ajayi.

**Writing – original draft:** Anthony Idowu Ajayi, Oluwaseyi Dolapo Somefun.

**Writing – review & editing:** Anthony Idowu Ajayi, Oluwaseyi Dolapo Somefun.

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
