## [Decision Letter · Decision Letter 0]

19 Feb 2020

PONE-D-19-31030

Recreational drug use among Nigerian university students: prevalence, determinants and frequency of use

PLOS ONE

Dear Dr Ajayi,

Thank you for submitting your manuscript to PLOS ONE. After careful consideration, we feel that it has merit but does not fully meet PLOS ONE’s publication criteria as it currently stands. Therefore, we invite you to submit a revised version of the manuscript that addresses the points raised during the review process.

We would appreciate receiving your revised manuscript by Apr 04 2020 11:59PM. To enhance the reproducibility of your results, we recommend that if applicable you deposit your laboratory protocols in protocols.io, where a protocol can be assigned its own identifier (DOI) such that it can be cited independently in the future. For instructions see: http://journals.plos.org/plosone/s/submission-guidelines#loc-laboratory-protocols

We look forward to receiving your revised manuscript.

Kind regards,

Aparup Das, Ph. D.

Academic Editor

PLOS ONE

Journal Requirements:

 https://doi.org/10.1017/S0021932018000330

In your revision ensure you cite all your sources (including your own works), and quote or rephrase any duplicated text outside the methods section. Additionally, in your Introduction and Discussion please cite and discuss your previous ONE publication and describe the current studies relationship to that manuscript. Ensure you cite any publications using the same dataset. Further consideration is dependent on these concerns being addressed.

3. We note you have included a table to which you do not refer in the text of your manuscript. Please ensure that you refer to Table 3 in your text; if accepted, production will need this reference to link the reader to the Table.

4. Your ethics statement must appear in the Methods section of your manuscript. If your ethics statement is written in any section besides the Methods, please move it to the Methods section and delete it from any other section. Please also ensure that your ethics statement is included in your manuscript, as the ethics section of your online submission will not be published alongside your manuscript.

Additional Editor Comments (if provided):

The manuscript has been reviewed by two independent reviewers and both have found the manuscript to be of merit for potential publication. However, they have indicated several issues which the authors need to address before the manuscript is re-considered. The authors are accordingly advised to revise the manuscript as suggested by the reviewers.  A table consisting of point-wise replies/corrections made in the manuscript (with page and line number) independently for each reviewer should be submitted along with the revised manuscript.

Reviewers' comments:

Reviewer's Responses to Questions

**Comments to the Author**

1. Is the manuscript technically sound, and do the data support the conclusions?

Reviewer #1: Partly

Reviewer #2: Yes

2. Has the statistical analysis been performed appropriately and rigorously? 

Reviewer #1: Yes

Reviewer #2: Yes

3. Have the authors made all data underlying the findings in their manuscript fully available?

Reviewer #1: Yes

Reviewer #2: Yes

4. Is the manuscript presented in an intelligible fashion and written in standard English?

Reviewer #1: Yes

Reviewer #2: Yes

5. Review Comments to the Author

Reviewer #1: This manuscript is about the recreational drug use and associated factors among Nigerian university students and has the potential to contribute to the literature if revised. The introduction is lengthy and organization could be improved. The stated theory guiding the analysis is only mentioned in the introduction and results are sometimes overstated given the data. Below are suggestions for revision.

Abstract

• The authors stated that they elicit determinants of recreational drug use, but it is a cross-sectional study so authors should be careful with using causal language.

Background

• Authors state abuse of opioids is growing in developed and more recently developing nations, yet the citations for developing nations are older than the citations for developed nations. Thus, it does not appear to be more recent in developing nations.

• It is unclear what the study is about until the very end of the background. Since the first sentence is about opioids, it appeared that that’s what the manuscript is about. Yet almost all of the studies cited do not report these substances, and the study measures codeine, tramadol, and marijuana together.

• The introduction can be better organized. Since SEM is used as the theoretical framework, that should be introduced earlier and the introduction follow that as the outline for drug use among young adults. That should lead to a paragraph before the hypotheses about what this manuscript contributes to the literature that we do not already know, and the purpose of the study.

Method

• Why were codeine, marijuana, and Tramadol combined into 1 question? The authors should explain why they did not assess substances separately.

• How is “ease tension” defined?

• Independent variables: More information is needed on each of the variables. How was age categorized? It is mentioned that family support is monetary and emotional. How was the question phrased to students? Did they understand the emphasis was on emotional support? And what family is included in family support? Was it specified that it was parents? Religious background was not included.

• It is unclear how SEM is applied in the methodology; authors should consider how the variables fit into the theoretical framework.

Discussion

• The authors state their study provides data to help address the opioid crisis, yet marijuana was included in the question so there is no way to know how many students actually used codeine or tramadol. The authors portray throughout the manuscript that this assesses opioid use, yet there is no way to know how many students actually used opioids. As such, the emphasis should be taken off of opioids and state recreational drug use instead.

• Some of the conclusions made are beyond the scope of the study. The authors mention mothers may be the pillar of support but that is unknown since who the support is provided by is not measured (just family). The authors should carefully consider all of their conclusions and ensure they are supported by results.

• The authors should consider other limitations mentioned in previous statements.

• There is no mention of SEM in the discussion and how it is applied to the results.

Reviewer #2: The manuscript fills in critical gap about the exposure of Nigerian university students to recreational drug use, and the finding about the protective role of mother in this scenario is enlightening. Still, few things remain unanswered in the paper like whether these mothers are working/non-working, what is the age range of participants, in which courses the students were enrolled (undergraduate/postgraduate) because that will predict their capacity to handle the enticements for drug use. Similarly, type of polygamy needs to be clarified. According to the manuscript when either parents are not alive then there is greater tendency to use recreational drugs, and this could be explored further more rigorously later. The United Nations Office on Drug Use and Crime came up with "Drug Use in Nigeria, 2018" which has elaborately discussed about use of recreational drug use and needs to be referred to by the authors for future studies. It is suggested to make the language of the manuscript crisper and more palatable and freer from typographical errors.

I wish the authors all the best for their future endeavors.

6. PLOS authors have the option to publish the peer review history of their article (what does this mean?). If published, this will include your full peer review and any attached files.

Reviewer #1: No

Reviewer #2: 

---

## [Author Response · Author response to Decision Letter 0]

19 Mar 2020

Reviewer #1: This manuscript is about the recreational drug use and associated factors among Nigerian university students and has the potential to contribute to the literature if revised. The introduction is lengthy and organization could be improved. The stated theory guiding the analysis is only mentioned in the introduction and results are sometimes overstated given the data. Below are suggestions for revision.

Response 

We thank the reviewer for these constructive comments. We have revised the manuscript accordingly. We have discussed the theory in the study discussion

Abstract

• The authors stated that they elicit determinants of recreational drug use, but it is a cross-sectional study so authors should be careful with using causal language.

Response: we have revised the manuscript and ensured all the use of determinants were changed to associated factors.

Background

• Authors state abuse of opioids is growing in developed and more recently developing nations, yet the citations for developing nations are older than the citations for developed nations. Thus, it does not appear to be more recent in developing nations.

Response: We have updated references to show more recent studies for developing countries. 

• It is unclear what the study is about until the very end of the background. Since the first sentence is about opioids, it appeared that that’s what the manuscript is about. Yet almost all of the studies cited do not report these substances, and the study measures codeine, tramadol, and marijuana together.

Response: The background which contains review of literature focuses on our measures of illicit drugs and this has been better highlighted. 

• The introduction can be better organized. Since SEM is used as the theoretical framework, that should be introduced earlier and the introduction follow that as the outline for drug use among young adults. That should lead to a paragraph before the hypotheses about what this manuscript contributes to the literature that we do not already know, and the purpose of the study.

Response: the introduction has been revamped. 

Method

• Why were codeine, marijuana, and Tramadol combined into 1 question? The authors should explain why they did not assess substances separately.

Response: we thank the reviewer for the insights and suggestions. These comments are cogent and have helped us to improve our manuscript. We performed a secondary analysis of dataset that only asked questions about recreational drug use. The question was phrased to probe use of drugs for recreational purposes. The questions gave examples of drugs such as codeine, marijuana and tramadol among others. Again the framing of the question contains examples of drugs people use in the study area for recreational purpose. This is an obvious limitation of this study and we have indicated this in the study limitation. 

• How is “ease tension” defined?

Response: We left the interpretation of stress and tension to our respondent because we did not consider it to be an ambiguous term. Also, participants in the pilot study did not find the meaning cumbersome. 

Response: 

• Independent variables: More information is needed on each of the variables. How was age categorized? It is mentioned that family support is monetary and emotional. How was the question phrased to students? Did they understand the emphasis was on emotional support? And what family is included in family support? Was it specified that it was parents? 

Response: These comments are really important and we have provided a detailed description of how all variables were measured. 

Religious background was not included.

Response: we have now included religion. 

• It is unclear how SEM is applied in the methodology; authors should consider how the variables fit into the theoretical framework.

Response : this is an important comment. We have describe how this model fits our data and acknowledge the limitation of our data.

Discussion

• The authors state their study provides data to help address the opioid crisis, yet marijuana was included in the question so there is no way to know how many students actually used codeine or tramadol. The authors portray throughout the manuscript that this assesses opioid use, yet there is no way to know how many students actually used opioids. As such, the emphasis should be taken off of opioids and state recreational drug use instead.

Response: we have refrain from this kind of generalisation and focus only on recreational drug use which is the main focus of this study. We thank the reviewer for this important comment.

• Some of the conclusions made are beyond the scope of the study. The authors mention mothers may be the pillar of support but that is unknown since who the support is provided by is not measured (just family). The authors should carefully consider all of their conclusions and ensure they are supported by results.

Response: we have deleted this sentence.

• The authors should consider other limitations mentioned in previous statements.

Response: we have elaborated on the limitations of the study.

• There is no mention of SEM in the discussion and how it is applied to the results.

Response: We have discussed the theory in our discussion.

Reviewer #2: The manuscript fills in critical gap about the exposure of Nigerian university students to recreational drug use, and the finding about the protective role of mother in this scenario is enlightening. 

Response: we appreciate the reviewer for the positive feedback

Still, few things remain unanswered in the paper like whether these mothers are working/non-working, 

Response: we did not measure the employment status or income of the mothers. 

What is the age range of participants, in which courses the students were enrolled. (undergraduate/postgraduate) because that will predict their capacity to handle the enticements for drug use. 

Response: We have indicated the age range of the students (17-34 years). They are all undergraduate students selected from first year to fifth year of study, including from all faculties of study in the two universities. 

Similarly, type of polygamy needs to be clarified. According to the manuscript when either parents are not alive then there is greater tendency to use recreational drugs, and this could be explored further more rigorously later. 

Response: we agree with the reviewer that more studies are needed to understand this link. 

The United Nations Office on Drug Use and Crime came up with "Drug Use in Nigeria, 2018" which has elaborately discussed about use of recreational drug use and needs to be referred to by the authors for future studies. 

Response: we did reference this report in our paper. “Nigeria has been noted to be the current highest consumer of cannabis and amphetamine in Africa (The United Nations Office on Drugs and Crime (UNODC), 2018).”

It is suggested to make the language of the manuscript crisper and more palatable and freer from typographical errors.

Response: we have edited our manuscript and it reads well now.

I wish the authors all the best for their future endeavors.

Response: the reviewer raised important insights that have allowed us to revise and improve our manuscript. We thank the reviewer for the important insights.

---

## [Decision Letter · Decision Letter 1]

8 Apr 2020

PONE-D-19-31030R1

Recreational drug use among Nigerian university students: prevalence, correlates and frequency of use

PLOS ONE

Dear Dr Ajayi,

Thank you for submitting your manuscript to PLOS ONE. After careful consideration, we feel that it has merit but does not fully meet PLOS ONE’s publication criteria as it currently stands. Therefore, we invite you to submit a revised version of the manuscript that addresses the points raised during the review process.

I now have the comments of the two reviewers who have reviewed this manuscript in its original form. As you can see from their comments, both have indicated 'minor revision'. I therefor would ask you to revise the manuscript carefully based on individual comments put by both the reviewers for further consideration.

We would appreciate receiving your revised manuscript by May 23 2020 11:59PM. To enhance the reproducibility of your results, we recommend that if applicable you deposit your laboratory protocols in protocols.io, where a protocol can be assigned its own identifier (DOI) such that it can be cited independently in the future. For instructions see: http://journals.plos.org/plosone/s/submission-guidelines#loc-laboratory-protocols

We look forward to receiving your revised manuscript.

Kind regards,

Aparup Das, Ph. D.

Academic Editor

PLOS ONE

Additional Editor Comments (if provided):

Please see my comments above

Reviewers' comments:

Reviewer's Responses to Questions

**Comments to the Author**

1. If the authors have adequately addressed your comments raised in a previous round of review and you feel that this manuscript is now acceptable for publication, you may indicate that here to bypass the “Comments to the Author” section, enter your conflict of interest statement in the “Confidential to Editor” section, and submit your "Accept" recommendation.

Reviewer #1: All comments have been addressed

Reviewer #2: (No Response)

2. Is the manuscript technically sound, and do the data support the conclusions?

Reviewer #1: Yes

Reviewer #2: Partly

3. Has the statistical analysis been performed appropriately and rigorously? 

Reviewer #1: Yes

Reviewer #2: Yes

4. Have the authors made all data underlying the findings in their manuscript fully available?

Reviewer #1: Yes

Reviewer #2: Yes

5. Is the manuscript presented in an intelligible fashion and written in standard English?

Reviewer #1: No

Reviewer #2: Yes

6. Review Comments to the Author

Reviewer #1: Overall, this manuscript is much improved. There are still some aspects the authors should consider.

Authors should ensure that the manuscript is written in past-tense since they already conducted the study, that all acronyms are defined (i.e., SSA for sub-Saharan Africa), and should consider hiring a copy-editor.

The introduction could be further refined. The authors only need 1 or 2 paragraphs introducing recreational drug use and why it needs to be addressed among this population. Then they should introduce their theory/model and how those variables at each level are related to recreational drug use.

In the introduction, the authors introduce factors in the introduction related to SEM, yet do not evaluate all of those factors in analyses. If the factors remain in the introduction, it should be discussed in the limitations sections why they were not included.

What are the implications of these findings as it relates to SEM? Rather than just listing the results of the different variables, what is the implication of these results (or where does future research need to go) to adequately apply SEM to recreational drug use among university students. The authors may want to consider another model as the focus seems to be on demographic and family/household factors rather than factors from the various SEM levels.

Reviewer #2: The study advances the discourse of recreational drug use in the Nigerian context, especially for the Nigerian youth going to universities and identifies protective factors against drug use in this context. The study has potential for publication and these points must be considered for the same:

Using SEM the study tries to explore various factors correlated with use of recreational drugs. In the abstract though there is no mention of SEM. In this model, however, how authors have defined and differentiated between the social and environmental factors is not clear. Say for e.g. why religion is considered as an environmental factor here – are there any previous studies which take such notions into consideration and/or relate recreational drug use with religion? This is so because it is more of a socio-cultural construct than an environmental one. Then, in table no. 1 religion is given as a demographic factor and later as an environmental one. I feel fitting of the present study into SEM is merely a post-hoc exercise and I have reservations about considering religion as an environmental and not social factor in the present study. The authors should shed more light on this aspect.

In the present study while delineating SEM the authors describe that they have taken into consideration the structure of family and not monitoring of young ones (pg 5). Any particular reason for not considering monitoring as a potential factor.

There are few instances of conflict in the manuscript which need to be resolved while revising like – On page 4 the sentence “We focused ….. recreational purpose.” Appears while later they show association between the mother’s influence on taking the recreational drugs, and here they are of the view that university students are free from parental control and supervision. Similarly, on pg. 5 it says “However, the federal ….. in Nigeria.”, then how come the participants in the present study were getting it for use.

In continuation of the previous point, the authors can consider the place of indulging in recreational drug which has not been delineated. This could be an environmental factor in the study. This is important because had the place been home then family factor comes into play. But if the place is university then already there is no role of family/parents there and the peer group assumes importance.

About study design - Was the study tool i.e. self-designed questionnaire influenced by some already existing questionnaire? I feel it is better to use “gender” instead of “sex” as an Independent variable. There is a typo when dealing with Independent variable – “school fee” should be “university fee”.

Results - in the present study the way in which use of recreational drugs has been explored confounds the fact that whether the users use it for “pleasure” or “to ease tension/stress”. The two things could not and should not be mixed in a single question because then the motivation for using the recreational drug gets diffused and not expressed.

Further, the study does not provide explanations for various findings like – why there is lesser current drug use among the female students than the male ones. Similarly, what is the reason for “our study shows that the nuclear family structure was independently associated with a lower likelihood of drug use in the unadjusted model” (pg. 16).

Discussion section– There are speculative sentences like “it is possible that the prevalence of drug use has increased since the 1990s.” (pg. 15) even though literature and reports like "Drug use in Nigeria, 2018" by United Nations Office on Drug Use and Crime are there. Further, it is felt that it is alright to to relate with presence of mother in the household, but this could be possible in a nuclear family scenario also where mother is also present along with the father and children.

Lastly, it is advised to the authors to not go for overarching ambitious conclusions which are beyond the ambit of present study.

7. PLOS authors have the option to publish the peer review history of their article (what does this mean?). If published, this will include your full peer review and any attached files.

Reviewer #1: No

Reviewer #2: No

---

## [Author Response · Author response to Decision Letter 1]

24 Apr 2020

Reviewer #1: Overall, this manuscript is much improved. There are still some aspects the authors should consider.

Authors should ensure that the manuscript is written in past-tense since they already conducted the study, that all acronyms are defined (i.e., SSA for sub-Saharan Africa), and should consider hiring a copy-editor.

Response: we defined all acronyms and copy-edited the manuscript. 

The introduction could be further refined. The authors only need 1 or 2 paragraphs introducing recreational drug use and why it needs to be addressed among this population. Then they should introduce their theory/model and how those variables at each level are related to recreational drug use.

Response: We have limited the introduction to only three paragraphs and cut down on the theoretical framework given that the theory only helped us to understand the influence of some family factors on recreational drug use.

In the introduction, the authors introduce factors in the introduction related to SEM, yet do not evaluate all of those factors in analyses. If the factors remain in the introduction, it should be discussed in the limitations sections why they were not included.

Response: The theory did not inform the study design, rather we drew on the theory to explain our results regarding the role of the family on recreational drug use among adolescents and young adults. We have indicated this in the study, under the study limitation as well as under suggestions for future studies. We have also removed factors that were not considered in the analysis from the introduction, 

What are the implications of these findings as it relates to SEM? Rather than just listing the results of the different variables, what is the implication of these results (or where does future research need to go) to adequately apply SEM to recreational drug use among university students. The authors may want to consider another model as the focus seems to be on demographic and family/household factors rather than factors from the various SEM levels.

Response: We have added a paragraph in the discussion to note the implication of our findings as it related to SEM. Also, we highlighted in this section our use SEM and the limitations thereof.

Response:

Reviewer #2: The study advances the discourse of recreational drug use in the Nigerian context, especially for the Nigerian youth going to universities and identifies protective factors against drug use in this context. The study has potential for publication and these points must be considered for the same:

Response: We thank the reviewer for the positive feedback as well as the constructive criticism of our paper. We have learnt a lot through this process and our paper has improved because of these comments. 

Using SEM the study tries to explore various factors correlated with use of recreational drugs. In the abstract though there is no mention of SEM. 

Response: we have added a sentence in abstract to introduce SEM. 

In this model, however, how authors have defined and differentiated between the social and environmental factors is not clear. Say for e.g. why religion is considered as an environmental factor here – are there any previous studies which take such notions into consideration and/or relate recreational drug use with religion? This is so because it is more of a socio-cultural construct than an environmental one. Then, in table no. 1 religion is given as a demographic factor and later as an environmental one. I feel fitting of the present study into SEM is merely a post-hoc exercise and I have reservations about considering religion as an environmental and not social factor in the present study. The authors should shed more light on this aspect.

Response: We agree with the reviewer that religion is a social factor and not an environmental factor. We have made this change in the study. We have retitled Table 1 as well to reflect this change. 

In the present study while delineating SEM the authors describe that they have taken into consideration the structure of family and not monitoring of young ones (pg 5). Any particular reason for not considering monitoring as a potential factor.

Response: We did not collect data on parental monitoring. The structure of the family is an indication of the possible time for monitoring. Young people whose parents had died for instance will have less parental monitoring compared with those whose parents are still alive and living with them. 

There are few instances of conflict in the manuscript which need to be resolved while revising like – On page 4 the sentence “We focused ….. recreational purpose.” Appears while later they show association between the mother’s influence on taking the recreational drugs, and here they are of the view that university students are free from parental control and supervision. Similarly, on pg. 5 it says “However, the federal ….. in Nigeria.”, then how come the participants in the present study were getting it for use.

Response: we have deleted these contradictions. The influence of parents on the life of child is lifelong. Even when one’s parent is not there, the need to represent them and not disappoint them continue to influence young people’s behaviours, especially in the context of Nigeria where several parents are responsible for young people most of their life course. 

In continuation of the previous point, the authors can consider the place of indulging in recreational drug which has not been delineated. This could be an environmental factor in the study. This is important because had the place been home then family factor comes into play. But if the place is university then already there is no role of family/parents there and the peer group assumes importance.

Response: This is difficult to tell given that we did not ask our participants where they use drugs. We do agree with the reviewer that these would help underscore the environmental factor and who will have more influence. Also, we did not collect data on peer-influence. These are limitations of this study and we have stated this.

About study design - Was the study tool i.e. self-designed questionnaire influenced by some already existing questionnaire? I feel it is better to use “gender” instead of “sex” as an Independent variable. 

Response: we have changed sex to gender

There is a typo when dealing with Independent variable – “school fee” should be “university fee”.

Response: we have made this change

Results - in the present study the way in which use of recreational drugs has been explored confounds the fact that whether the users use it for “pleasure” or “to ease tension/stress”. The two things could not and should not be mixed in a single question because then the motivation for using the recreational drug gets diffused and not expressed.

Response: While we agree with the reviewer’s point, we will like to also note that one of the underlying reasons for using drugs for pleasure could be to improve life in some ways. Most people that use drugs says it helps to ease tension. These drugs are also often called recreational drugs. Recreational drugs is a loose term that refers to legal and illegal drugs that are used without medical supervision (https://www.bmj.com/content/353/bmj.i2775/rr). Our inclusion of tension in the question is to have a broad response from our participants given that the drugs themselves are referred to as recreational drugs. 

Further, the study does not provide explanations for various findings like – why there is lesser current drug use among the female students than the male ones. Similarly, what is the reason for “our study shows that the nuclear family structure was independently associated with a lower likelihood of drug use in the unadjusted model” (pg. 16).

Response: We have added these points in our discussion. Boys are more favourable to taking risks and society frown at girls drug use. 

Discussion section– There are speculative sentences like “it is possible that the prevalence of drug use has increased since the 1990s.” (pg. 15) even though literature and reports like "Drug use in Nigeria, 2018" by United Nations Office on Drug Use and Crime are there. 

Response: we have added references to these kinds of statements. 

Further, it is felt that it is alright to relate with presence of mother in the household, but this could be possible in a nuclear family scenario also where mother is also present along with the father and children. 

Response: We have explained our position on this based on the fact that emotional nurturance may be higher among mothers regardless of the family structure. It is also possible that mothers may have more appropriate consequences for youth behaviours which may eventually protect them. 

Lastly, it is advised to the authors to not go for overarching ambitious conclusions which are beyond the ambit of present study.

Response: we do agree with the reviewer and have noted this point in the discussion. We have revised the conclusion to underscore the reviewer’s point.

---

## [Editor Report · Decision Letter 2]

27 Apr 2020

Recreational drug use among Nigerian university students: prevalence, correlates and frequency of use

PONE-D-19-31030R2

Dear Dr. Ajayi,

We are pleased to inform you that your manuscript has been judged scientifically suitable for publication and will be formally accepted for publication once it complies with all outstanding technical requirements.

With kind regards,

Aparup Das, Ph. D.

Academic Editor

PLOS ONE

---

## [Editor Report · Acceptance letter]

29 Apr 2020

PONE-D-19-31030R2 

Recreational drug use among Nigerian university students: prevalence, correlates and frequency of use 

Dear Dr. Ajayi:

I am pleased to inform you that your manuscript has been deemed suitable for publication in PLOS ONE. Congratulations! Your manuscript is now with our production department. 

With kind regards,

on behalf of

Dr. Aparup Das 

Academic Editor

PLOS ONE